# Probing Representations for Document-level Event Extraction

**Barry Wang[1]** and **Xinya Du[2]** and **Claire Cardie[1]**

[1]Department of Computer Science, Cornell University

[2] Department of Computer Science, University of Texas at Dallas

zw545@cornell.edu, xinya.du@utdallas.edu, cardie@cs.cornell.edu

## Abstract

The probing classifiers framework has been employed for interpreting deep neural network models for a variety of natural language processing (NLP) applications. Studies, however, have largely focused on sentence-level NLP tasks. This work is the first to apply the probing paradigm to representations learned for document-level information extraction (IE). We designed eight embedding probes to analyze surface, semantic, and event-understanding capabilities relevant to document-level event extraction. We apply them to the representations acquired by learning models from three different LLM-based document-level IE approaches on a standard dataset. We found that trained encoders from these models yield embeddings that can modestly improve argument detections and labeling but only slightly enhance event-level tasks, albeit trade-offs in information helpful for coherence and event-type prediction. We further found that encoder models struggle with document length and cross-sentence discourse.

## 1 Introduction

Relation and event extraction (REE) focuses on identifying clusters of entities participating in a shared relation or event from unstructured text, that frequently contains a fluctuating number of such instances. While the field of information extraction (IE) started out building training and evaluation REE datasets primarily concerned with documents, researchers have been overwhelmingly focusing on sentence-level datasets (Li et al., 2013; Du and Cardie, 2020). Nevertheless, many IE tasks require a more comprehensive understanding that often extends to the entire input document, leading to challenges such as length and multiple events when embedding full documents. Consequently, document-level datasets continue to pose challenges for even the most advanced models today (Das et al., 2022).

REE is considered an essential and popular task, encompassing various variations. One particularly general approach is template filling[1], which can subsume certain other IE tasks by formatting. In this regard, our focus lies on template-extraction methods with an end-to-end training scheme, where texts serve as the sole input.

Multi-task NLP models often support and are evaluated on the task. As a result, we have seen frameworks of diverse underlying assumptions and architectures for the task. Nevertheless, high-performing modern models all leverage and fine-tune on pre-trained neural contextual embedding models, like variations of BERT (Devlin et al., 2019), due to the generalized performance leap introduced by transformers and pretraining.

It is crucial to understand these representations of the IE frameworks, as doing so reveals model strengths and weaknesses. However, unlike lookup style embeddings such as GloVe (Pennington et al., 2014), these neural contextualized representations are inherently difficult to interpret, leading to ongoing research efforts focused on analyzing their encoded information (Tenney et al., 2019b; Zhou and Srikumar, 2021; Belinkov, 2022). This work is inspired by various sentence-level embedding interpretability works, including Conneau et al. (2018) and Alt et al. (2020).

Yet, to the best of our knowledge, no prior work has been done to understand the embedding of features that exist only at the document-level scale. Hence, our work aims to fill this gap by investigating the factors contributing to the model performance. Specifically, we analyze the impact of three key elements: contextualization of encoding, fine-tuning, and encoder and post-encoding architectures. Our contributions can be summarized as follows:

---

[1]Defined in Appendix A. Template filling might not subsume certain relation extraction like n-ary relation extraction. Hence we will prefer "event extraction" in the following text.

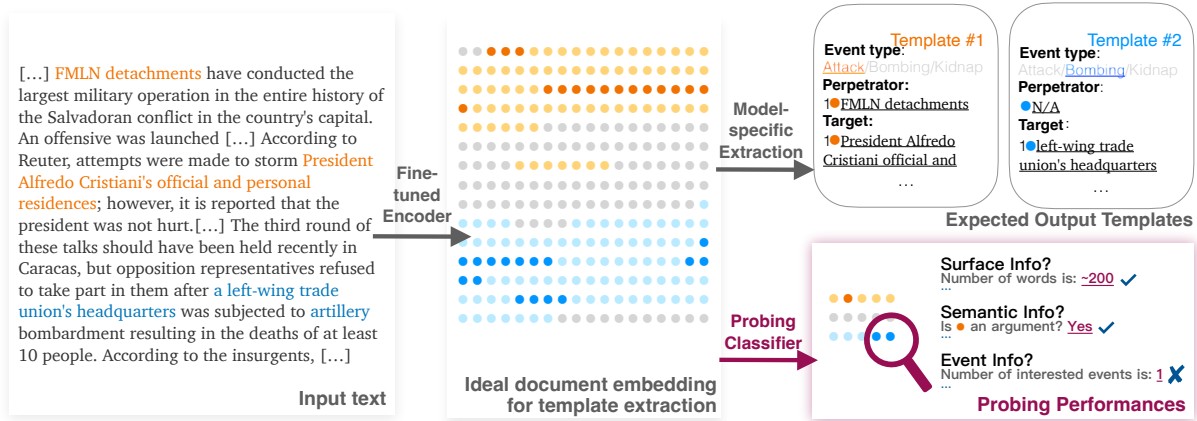

**Figure 1: Overview of an ideal event extraction example and probing.** Ideally, after contextualization by a fine-tuned encoder on the IE task, the per-token embedding can capture richer semantic and related event information, thereby facilitating an easier model-specific extraction process. Our probing tasks test how different frameworks and conditions (e.g. IE training, coherence information access) affect information captured by the embeddings.

- We identified the necessary document-level IE understanding capabilities and created a suite of probing tasks[2] corresponding to each.
- We present a fine-grained analysis of how these capabilities relate to encoder layers, full-text contextualization, and fine-tuning.
- We compare IE frameworks of different input and training schemes and discuss how architectural choices affect model performances.

## 2 Probing and Probing Tasks

The ideal learned embedding for spans should include features and patterns (or generally, "information") that independently show similarity to other span embeddings of the same entity mentions, in the same event, etc., and we set out to test if that happens for trained encoders.

Probing uses simplified tasks and classifiers to understand what information is encoded in the embedding of the input texts. We train a given document-level IE model (which finetunes its encoder in training), and at test time capture the output of its encoder (the document representations) before they are further used in the model-specific extraction process. We then train and run our probing tasks, each assessing an encoding capability of the encoder.

Drawing inspiration from many sentence-level probing works, we adopt some established setups and tasks, but with an emphasis on probing tasks pertaining to document and event understanding.

We use the MUC document-level IE dataset

| Category | Illustration | Task | Task Full Name |
|---|---|---|---|
| Surface | ● ● ... ● −>#Words | WordCt | Word Count |
| | ● ● ... ● −>#Sent−ences | SentCt | Sentence Count |
| Semantic | ● a.k.a. ● ? | Coref | Are Coreferent |
| | ● in [Any] ? | IsArg | Is an Argument |
| | ● −> Perpetrator? Victim? ... | ArgTyp | Argument Type |
| Event | ● −> Bombing/ Attack... ? | EnvtTyp₂ | Event Type |
| | ● both in [Any] ? | CoEnvt | Co-Event |
| | ● ● ... ● −># | EnvtCt | Event Count |

**Figure 2: Probing Task Illustrations.** Each ● refers to a span embedding (which is an embedding of a token in our experiment), and non-gray ● means embeddings are known to be a role filler. See Section 2 for full descriptions.

(muc, 1991) as our base dataset (details in Section 3.3) to develop evaluation probing tasks.

We present our probing tasks in Figure 2, This section outlines the probing tasks used for assessing the effectiveness of the learned document representations.

When designing these probing tasks, our goal is to ensure that each task accurately measures a specific and narrow capability, which was often a subtask in traditional pipelined models. Additionally, we want to ensure fairness and generalizability in these tasks. Therefore, we avoid using event triggers in our probing tasks, especially considering that not all models use them during training.

We divide our probing tasks into three cate-

---

[2]Our model and probing codea are publicly available at
https://github.com/GithuBarry/DocIE-Probing.

gories: surface information, generic semantic understanding, and event understanding.

**Surface information** These tasks assess if text embeddings encode the basic surface characteristics of the document they represent. Similar to the sentence length task proposed by Adi et al. (2017), we employed a word count (**WordCt**) task and a sentence count (**SentCt**) task, each predicts the number of words and sentences in the text respectively. Labels are grouped into 10 count-based buckets and ensured a uniform distribution.

**Semantic information** These tasks go beyond surface and syntax, capturing the conveyed meaning between sentences for higher-level understanding. Coreference (**Coref**) is the binary-classification task to determine if the embeddings of two spans of tokens ("mentions") refer to the same entity. Due to the annotation of MUC, all used embeddings are all known role-fillers, which are strictly necessary for the downstream document-level IE to avoid duplicates. To handle varying mention span lengths in MUC, we utilize the first token's embedding for effective probing classifier training, avoiding insufficient probe training at later positions. A similar setup applies to role-filler detection (**IsArg**), which predicts if a span embedding is an argument of any template. This task parallels Argument Detection in classical models. Furthermore, the role classification task (**ArgTyp**) involves predicting the argument type of the role-filler span embedding. This task corresponds to the argument extraction (argumentation) step in classical pipelines.

**Event understanding** The highest level of document-level understanding is event understanding. To test the model's capability in detecting events, we used an event count task (**EvntCt**) where the probing classifier is given the full-text embedding and asked to predict the number of events that occurred in the text. We split all count labels into three buckets for class balancing. To understand how word embeddings are helpful to event deduplication or in argument linking, our Co-event task (**CoEvnt**) takes two argument span embeddings and predicts whether they are arguments to the same event or different ones. Additionally, the event type task (**EvntTyp**$_n$) involves predicting the type of the event template based on the embeddings of $n$ role filler first tokens. This task is similar to the classical event typing subtask, which often uses triggers as inputs. By performing this task, we can assess whether fine-tuning makes event-type information explicit.

Although syntactic information is commonly used in probing tasks, document-level datasets have limited syntactic annotations due to the challenges of accurately annotating details like tree-depth data at scale. While the absence of these tasks is not ideal, we believe it would not significantly impact our overall analysis.

## 3 Experiment Setup

### 3.1 IE Frameworks

We train the following document-level IE frameworks for $5, 10, 15, 20$ epochs on MUC, and we observe the lowest validation loss or highest event F1 score at epoch 20 for all these models.

**DyGIE++ (Wadden et al., 2019)** is a framework capable of named entity recognition, relation extraction, and event extraction tasks. It achieves all tasks by enumerating and scoring sections (spans) of encoded text and using the relations of different spans to detect triggers and construct event outputs.

**GTT (Du et al., 2021)** is a sequence-to-sequence event-extraction model that perform the task end-to-end, without the need of labeled triggers. It is trained to decode a serialized template, with tuned decoder constraints.

**TANL (Paolini et al., 2021)** is a multi-task sequence-to-sequence model that fine-tunes T5 model (Raffel et al., 2020) to translate text input to augmented natural languages, with the in-text augmented parts extracted to be triggers and roles. It uses a two stage approach for event extraction, by first decoding (translating) the input text to extract trigger detection, then decoding related arguments for each trigger predicted.

### 3.2 Probing Model

We use a similar setup to SentEval (Conneau et al., 2018) with an extra layer. While sentence-level probing can use all dimensions of embeddings as input, we added an attention-weighted layer right after the input layer, as to simulate a response to a trained query and to reduce dimensions. The 768-dimension layer-output is then trained using

| Model (IE-F1) | Input | WordCt | SentCt | IsArg | ArgTyp | Coref | EvntTyp$_2$ | CoEvnt | EvntCt | Avg |
|---|---|---|---|---|---|---|---|---|---|---|
| **DyGIE++** | FullText | 58.6 | 47.0 | 87.1 | 83.8 | 64.7 | **60.5** | 73.6 | 67.2 | 67.8 |
| (41.9) | SentCat | 57.4 | 58.9 | 87.5 | 85.6 | 69.2 | 56.7 | 67.9 | 67.0 | 68.8 |
| **GTT** | FullText | 58.6 | 46.3 | 88.3 | **88.5** | 66.7 | 60.4 | 66.4 | **68.3** | 67.9 |
| (49.0) | SentCat | 55.8 | **58.9** | 88.6 | 88.0 | 69.5 | 57.5 | 65.07 | 67.5 | 68.8 |
| **TANL** | FullText | 54.2 | 43.3 | 88.2 | 86.8 | 66.6 | 57.8 | 60.0 | 65.8 | 65.3 |
| (33.2) | SentCat | 34.3 | 40.8 | 88.2 | 87.0 | 65.6 | 53.5 | 59.8 | 67.0 | 62.0 |
| **BERT**$_{base}$ | FullText | **65.5** | 45.0 | 87.8 | 86.1 | **75.7** | 60.4 | **74.0** | 63.5 | 69.7 |

**Table 1: Probing Task Test Average Accuracy.** IE frameworks trained for 20 epochs on MUC, and we run probing tasks on the input representations. We compare the 5-trial averaged test accuracy on full-text embeddings and concatenation of sentence embeddings from the same encoder to the untrained BERT baseline. IE-F1 refers to the model's F1 score on MUC test. Underlined data are the best in same embedding method, while bold, overall. We further report data over more epochs in Table 7, and results on **WikiEvents** in Table 8 in Appendix E.

the same structure as SentEval. Specific training detail can be found in Appendix D.

### 3.3 Dataset

We use MUC-3 and MUC-4 as our document-level data source to create probing tasks, thanks to its rich coreference information. The dataset has 1300/200/200 training/validation/testing documents. any dataset with a similar format can be used to create probing tasks as well, and we additionally report results on the smaller WikiEvent (Li et al., 2021) Dataset in table 8 in Appendix E. More MUC descriptions available in Appendix B.

## 4 Result and Analysis

We present our data in Table 1, with results in more epochs available in Table 7 in Appendix E.

**Document-level IE Training and Embeddings** Figure 3 shows that embedded semantic and event information fluctuate during IE training, but steadily differ from the untrained BERT-base baseline. For the document representation, trained encoders significantly enhance embeddings for event detection as suggested by the higher accuracy in event count predictions (EvntCt↑). At the span level, embeddings lose information crucial for event type prediction and coreference, as evidenced by decreased event typing performance (EvntTyp$_2$↓) and coreference accuracy (Coref↓) over IE training epochs. Note again that coreference data pairs used are role-fillers and hence crucial for avoiding duplicated role-extractions, and future frameworks could seek to lower this knowledge loss. Nevertheless, IE training does aid argument detection (IsArg↑) and role labeling

(ArgTyp↑), albeit less consistently.

| Model | FullText Best | FullText Avg | Sent Best | Sent Avg |
|---|---|---|---|---|
| WordCount: ≤ 209 | | | | |
| DyGIE++ | 68.5 | 67.1 | **69.7** | **68.8** |
| GTT | 70.3 | **68.7** | **72.1** | 68.0 |
| TANL | **71.8** | **70.2** | 66.3 | 64.2 |
| WordCount: 210-420 | | | | |
| DyGIE++ | 67.0 | **65.7** | **67.6** | 64.7 |
| GTT | **67.6** | **67.0** | 66.4 | 64.7 |
| TANL | **64.8** | **62.0** | 63.6 | 60.8 |
| WordCount: ≥ 431 | | | | |
| DyGIE++ | 70.6 | 70.2 | **74.2** | **72.1** |
| GTT | 69.1 | 68.7 | **71.5** | **70.2** |
| TANL | 67.3 | 65.2 | **69.7** | **68.3** |

**Table 2: EvntCt Probing Test Accuracy (%)** 5 random seed averaged. When WordCount ≥431, both FullText and SentCat embeddings are truncated to the same length (e.g., BERT-base has a limit of 512) for comparison fairness. Concatenated sentence embeddings show an advantage on medium or long texts.

**Probing Performance of Different Models** Table 1 highlights the strengths and weaknesses of encoders trained using different IE frameworks. In addition to above observations, we see that DyGIE++ and GTT document embeddings capture event information (EvntCt↑) only marginally better than the baseline, whereas the TANL-finetuned encoder often has subpar performance across tasks. This discrepancy may be attributed to TANL's usage of T5 instead of BERT, which might be more suitable for the task, and that TANL employs the encoder only once but the decoder multiple times, resulting in less direct weight updates for the encoder and consequently lower its perfor-

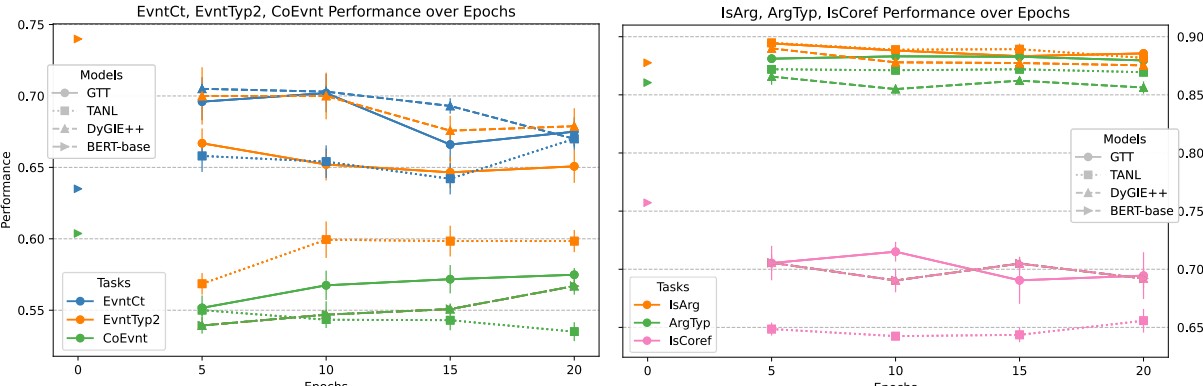

**Figure 3: Probing accuracy on event (left) and semantic (right) information over document-level IE training epoch**. 5 random seed results averaged (with standard deviation error bars). Color-coded by probing tasks. Trained encoder gain and lose information in their generated embeddings as they are trained for the IE tasks.

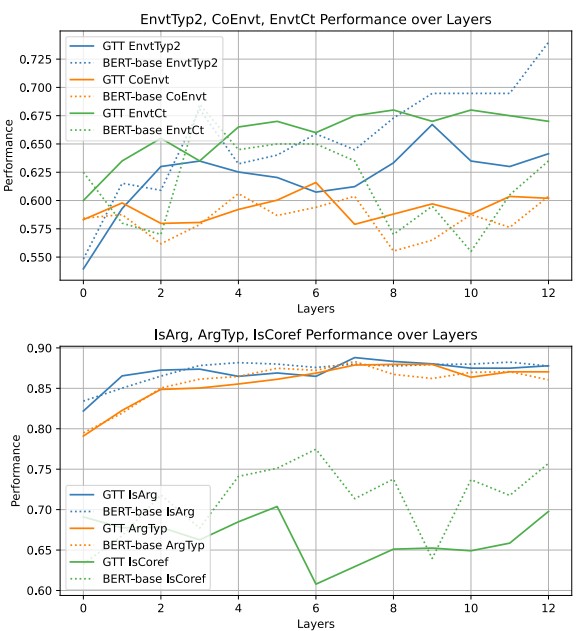

**Figure 4: Probing accuracy on event (upper) and semantic (lower) information over encoder layers** from GTT trained over 18 epoch and BERT-base.

mance in probing tasks (and the document-level IE task itself). Surface information encoding (Figure 6 in Appendix E) differ significantly by models.

**Sentence and Full Text Embedding** As demonstrated in Table 1, embedding sentences individually and then concatenating them can be more effective for IE tasks than using embeddings directly from a fine-tuned encoder designed for entire documents. Notably, contextually encoding the full text often results in diminished performance in argument detection (IsArg↓), labeling (ArgTyp↓), and particularly in Event detection (EvntCt↓) for shorter texts, as highlighted in Table 2. These results suggest that encoders like BERT might not effectively utilize cross-sentence

discourse information, and a scheme that can do so remains an open problem. However, contextualized embedding with access to the full text does encode more event information in its output representation for spans (CoEvnt↑).

**Encoding layers** Lastly, we experiment to locate the encoding of IE information in different layers of the encoders, a common topic in previous works (Tenney et al., 2019a). Using GTT with the same hyperparameter in its publication, its finetuned encoder shows semantic information encoding mostly (0-indexed) up to layer 7 (IsArg↑, ArgTyp↑), meanwhile, event detection capability increases throughout the encoder (CoEvnt↑, EvntCt↑). Surface information (Figure 5 in Appendix E) generally remains the same.

## 5 Conclusion

Our work pioneers the application of probing to the representation used at the document level, specifically in event extraction. We observed semantic and event-related information embedded in representations varied throughout IE training. While encoding improves on capabilities like event detection and argument labeling, training often compromises embedded coreference and event typing information. Comparisons of IE frameworks uncovered that current models marginally outperformed the baseline in capturing event information at best. Our analysis also suggested a potential shortcoming of encoders like BERT in utilizing cross-sentence discourse information effectively. In summary, our work provides the first insights into document-level representations, suggesting new research directions for optimizing these representations for event extraction tasks.

## Acknowledgements

We would like to express our gratitude to the following undergraduate contributors who played vital roles in this research:

Maitreyi Chatterjee, for her diligent efforts in exploring the MUC dataset, experimenting with contextual word embeddings, and making valuable contributions to the appendix.

Wayne Chen, whose contributions were indispensible in adapting TANL for the generic document-level IE task.

## Limitations

**Dataset**   While other document-level IE datasets are possible, none of them offer rich details like MUC. For example, document-level n_ary_relations datasets like SciREX(Jain et al., 2020) can only cover three out of the six semantic and event knowledge probing tasks, and the dataset has issues with missing data.

Additionally, we focus on template-filling-capable IE frameworks as they show more generality in applications (and is supported by more available models like GTT), barring classical relation extraction task dataset like the DocRED(Yao et al., 2019).

**Scoping**   While we observe ways to improve document-level IE frameworks, creating new frameworks and testing them are beyond the scope of this probing work.

**Embedding length and tokenizer**   All models we investigated use an encoder that has an input cap of 512 tokens, leaving many entities inaccessible. In addition, some models use tokenizers that tokenize words into fewer tokens and as a result, may access more content in full-text embedding probing tasks. Note that also because of tokenizer difference, despite our effort to make sure all probing tasks are fair, some models might not see up to 2.1% training data while others do.

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

# A   Definition of Template Filling

Assume a predefined set of event types, $T_1, ..., T_m$, where $m$ represents the total number of template types. Every event template comprises a set of $k$ roles, depicted as $r_1, ..., r_k$. For a document made up of $n$ words, represented by $x_1, x_2, ..., x_n$, the template filling task is to extract zero or more templates. The number of templates are not given as an input, and each template can represent a n-ary relation or an event.

Each extracted template contains $k + 1$ slots: the first slot is dedicated to the event type, which is one of the event types from $T_1, ..., T_m$. The subsequent $k$ slots represent an event role, which will be one of the roles $r_1, ..., r_k$. The system's job is to assign zero or more entities (role-fillers) to the corresponding role in each slot.

# B   MUC dataset

The MUC 3 dataset (1991) comprises news articles and documents manually annotated for coreference resolution and for resolving ambiguous references in the text. The MUC 4 dataset(1992), on the other hand, expanded the scope to include named entity recognition and template-based information extraction.

We used a portion of the MUC 3 and 4 datasets for template filling and labeled the dataset with triggers based on event types for our probing tasks. The triggers were added to make the dataset compatible with TANL so that we could perform multi-template prediction.

The event schema includes 5 incident types - namely 'kidnapping', 'attack', 'bombing', 'robbery', 'forced work stoppage', and 'arson'. The coreference information for each event includes fields like 'PerpInd', 'PerpOrg', 'Target', 'Victim', and 'Weapon'.

## C IE Framework Parameters

See Table 3, 4, 5

| Parameter | Value |
|---|---|
| num_epochs | [5, 10, 15, 20] |
| patience | 8 |
| max_span_width | 8 |
| optimizer + | lr: 5e-4 |
| bert_model | bert-base-uncased |
| target_task | events |

Table 3: DyGIE++ Model Parameters

| Parameter | Value |
|---|---|
| –max_seq_length_src | 435 |
| –max_seq_length_tgt | 75 |
| –num_train_epochs | [5, 10, 15, 18, 20] |
| –bert_model | bert-base-uncased |
| –thresh | 80 |
| –batch_size | 1 |

Table 4: GTT Model Parameters

| Parameter | Value |
|---|---|
| multitask | True |
| model_name_or_path | t5-base |
| num_train_epochs | [5, 10, 15, 20] |
| tokenizer_name | t5-base |
| max_seq_length | 512 |
| max_seq_length_eval | 512 |
| per_device_train_batch_size | 4 |
| per_device_eval_batch_size | 1 |
| num_beams | 1 |

Table 5: TANL Model Training Parameters

## D Probing Model Details

See Table 6.

## E Additional Results

See Figure 5 and Figure 6 for more probing results on MUC.

See Table 8 for more results on **WikiEvents**. WikiEvents is a smaller (246-example) dataset.

| Parameter | Value |
|---|---|
| nhid | 400, (100, 200, 800) |
| tenacity | 10 |
| batch_size | 8 |
| MaxEpoch | 1000 |
| optim | adam |
| dropout | 0, (0.1) |
| attention-head | 1, (11, 22) |

Table 6: **Probing model parameters** values in the parenthesis are tested by not used, often due to lower performances.

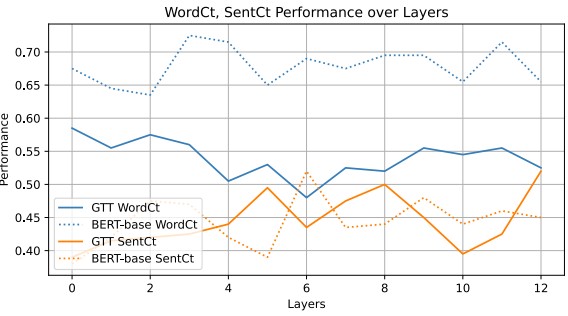

Figure 5: **Probing accuracy on semantic surface information over encoder layers** from GTT trained over 18 epochs and BERT-base.

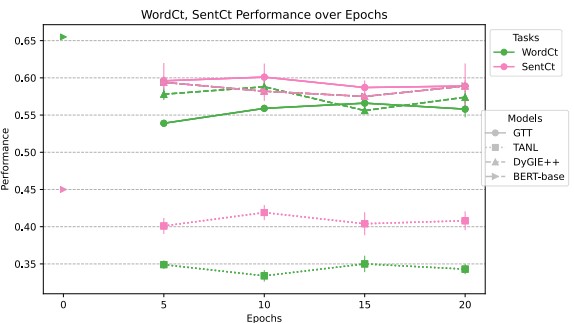

Figure 6: **Probing accuracy on surface information** over document-level IE training epoch.

| Model | Epoch | Embedding | WordCt | SentCt | IsArg | ArgTyp | Coref | CoEvnt | EvntTyp$_2$ | EvntCt |
|---|---|---|---|---|---|---|---|---|---|---|
| GTT | 5 | FullText | $59.50_{\pm4.65}$ | $45.10_{\pm2.66}$ | $89.37_{\pm1.02}$ | $87.58_{\pm0.33}$ | $71.05_{\pm1.59}$ | $59.54_{\pm0.98}$ | $67.56_{\pm2.84}$ | $68.50_{\pm2.21}$ |
| | | SentCat | $53.90_{\pm0.65}$ | $59.60_{\pm5.98}$ | $89.42_{\pm0.70}$ | $88.11_{\pm0.35}$ | $70.53_{\pm3.70}$ | $55.16_{\pm2.17}$ | $66.69_{\pm2.59}$ | $69.60_{\pm3.23}$ |
| | 10 | FullText | $59.30_{\pm3.37}$ | $47.10_{\pm3.90}$ | $88.68_{\pm1.04}$ | $87.56_{\pm0.49}$ | $67.06_{\pm3.65}$ | $61.41_{\pm2.01}$ | $65.14_{\pm0.89}$ | $68.40_{\pm2.19}$ |
| | | SentCat | $55.90_{\pm1.24}$ | $60.10_{\pm4.56}$ | $88.81_{\pm1.02}$ | $88.31_{\pm0.62}$ | $71.51_{\pm2.13}$ | $56.74_{\pm2.57}$ | $65.20_{\pm2.79}$ | $70.20_{\pm3.35}$ |
| | 15 | FullText | $59.60_{\pm4.89}$ | $44.90_{\pm2.51}$ | $87.99_{\pm0.58}$ | $88.44_{\pm0.42}$ | $67.75_{\pm1.56}$ | $61.32_{\pm1.23}$ | $66.46_{\pm1.59}$ | $68.40_{\pm2.61}$ |
| | | SentCat | $56.60_{\pm0.89}$ | $58.70_{\pm1.35}$ | $88.33_{\pm0.80}$ | $88.28_{\pm0.74}$ | $69.05_{\pm5.05}$ | $57.17_{\pm2.48}$ | $64.65_{\pm2.78}$ | $66.60_{\pm2.92}$ |
| | 20 | FullText | $58.60_{\pm1.95}$ | $46.30_{\pm2.93}$ | $88.31_{\pm0.90}$ | $88.51_{\pm0.83}$ | $66.68_{\pm1.91}$ | $60.43_{\pm0.87}$ | $66.40_{\pm2.23}$ | $68.30_{\pm1.82}$ |
| | | SentCat | $55.80_{\pm2.77}$ | $58.90_{\pm1.92}$ | $88.56_{\pm0.34}$ | $87.96_{\pm0.96}$ | $69.45_{\pm5.04}$ | $57.48_{\pm1.18}$ | $65.07_{\pm2.89}$ | $67.50_{\pm2.47}$ |
| TANL | 5 | FullText | $55.70_{\pm2.25}$ | $44.50_{\pm2.06}$ | $89.65_{\pm0.32}$ | $87.39_{\pm0.99}$ | $65.49_{\pm1.29}$ | $57.65_{\pm0.91}$ | $58.81_{\pm1.45}$ | $67.30_{\pm2.97}$ |
| | | SentCat | $34.90_{\pm1.43}$ | $40.10_{\pm2.68}$ | $89.47_{\pm0.46}$ | $87.20_{\pm0.46}$ | $64.86_{\pm1.39}$ | $55.01_{\pm1.02}$ | $56.85_{\pm1.88}$ | $65.80_{\pm2.80}$ |
| | 10 | FullText | $54.20_{\pm1.52}$ | $41.30_{\pm4.40}$ | $88.89_{\pm0.52}$ | $86.90_{\pm0.54}$ | $63.63_{\pm1.39}$ | $56.88_{\pm1.74}$ | $62.06_{\pm1.45}$ | $66.50_{\pm1.77}$ |
| | | SentCat | $33.40_{\pm1.92}$ | $41.90_{\pm2.53}$ | $88.88_{\pm0.61}$ | $87.12_{\pm0.39}$ | $64.25_{\pm0.84}$ | $54.33_{\pm0.48}$ | $59.94_{\pm3.19}$ | $65.40_{\pm2.86}$ |
| | 15 | FullText | $52.10_{\pm1.47}$ | $43.30_{\pm2.25}$ | $88.79_{\pm1.08}$ | $87.08_{\pm0.70}$ | $67.32_{\pm1.97}$ | $56.70_{\pm0.98}$ | $60.32_{\pm3.15}$ | $64.80_{\pm1.04}$ |
| | | SentCat | $35.00_{\pm2.76}$ | $40.40_{\pm3.86}$ | $88.93_{\pm1.16}$ | $87.20_{\pm0.30}$ | $64.36_{\pm1.56}$ | $54.30_{\pm1.77}$ | $59.84_{\pm2.69}$ | $64.20_{\pm2.73}$ |
| | 20 | FullText | $54.20_{\pm1.48}$ | $43.30_{\pm1.60}$ | $88.15_{\pm0.53}$ | $86.81_{\pm0.60}$ | $66.62_{\pm1.85}$ | $57.77_{\pm1.05}$ | $60.03_{\pm1.46}$ | $65.80_{\pm2.73}$ |
| | | SentCat | $34.30_{\pm1.68}$ | $40.80_{\pm3.17}$ | $88.17_{\pm0.68}$ | $86.95_{\pm0.36}$ | $65.57_{\pm2.54}$ | $53.50_{\pm1.66}$ | $59.84_{\pm1.94}$ | $67.00_{\pm1.50}$ |
| DyGIE++ | 5 | FullText | $58.10_{\pm2.43}$ | $51.80_{\pm5.90}$ | $89.06_{\pm0.50}$ | $87.43_{\pm1.03}$ | $64.26_{\pm5.98}$ | $57.90_{\pm1.72}$ | $73.43_{\pm3.57}$ | $68.70_{\pm1.96}$ |
| | | SentCat | $57.80_{\pm1.96}$ | $59.40_{\pm2.07}$ | $88.99_{\pm0.62}$ | $86.57_{\pm1.76}$ | $70.56_{\pm1.78}$ | $53.93_{\pm1.43}$ | $70.00_{\pm5.01}$ | $70.50_{\pm2.03}$ |
| | 10 | FullText | $55.80_{\pm6.02}$ | $47.10_{\pm2.41}$ | $88.18_{\pm0.87}$ | $84.87_{\pm0.80}$ | $63.64_{\pm6.56}$ | $57.71_{\pm3.56}$ | $72.15_{\pm3.56}$ | $67.80_{\pm1.68}$ |
| | | SentCat | $58.80_{\pm2.02}$ | $58.20_{\pm3.25}$ | $87.80_{\pm0.65}$ | $85.50_{\pm1.01}$ | $69.04_{\pm2.55}$ | $54.69_{\pm2.34}$ | $70.00_{\pm4.09}$ | $70.30_{\pm1.15}$ |
| | 15 | FullText | $60.20_{\pm3.17}$ | $48.70_{\pm5.77}$ | $87.93_{\pm0.71}$ | $84.07_{\pm1.38}$ | $66.27_{\pm3.83}$ | $60.68_{\pm2.30}$ | $72.55_{\pm3.70}$ | $68.30_{\pm1.20}$ |
| | | SentCat | $55.60_{\pm0.82}$ | $57.50_{\pm5.39}$ | $87.74_{\pm0.87}$ | $86.22_{\pm0.85}$ | $70.49_{\pm1.43}$ | $55.08_{\pm0.99}$ | $67.57_{\pm2.61}$ | $69.30_{\pm1.35}$ |
| | 20 | FullText | $58.60_{\pm5.37}$ | $47.00_{\pm5.67}$ | $87.13_{\pm0.50}$ | $83.83_{\pm1.21}$ | $64.65_{\pm7.17}$ | $60.50_{\pm1.57}$ | $73.58_{\pm2.66}$ | $67.20_{\pm1.64}$ |
| | | SentCat | $57.40_{\pm2.10}$ | $58.90_{\pm7.59}$ | $87.53_{\pm0.55}$ | $85.63_{\pm1.32}$ | $69.20_{\pm2.09}$ | $56.69_{\pm1.50}$ | $67.88_{\pm3.14}$ | $67.00_{\pm1.94}$ |
| BERT$_{base}$ | | FullText | **65.50** | 45.00 | 87.76 | 86.05 | **75.72** | 60.37 | **73.99** | 63.50 |

**Table 7: Probing test accuracy on more epoch.** Note that underlined data, unlike those presented in Table 1, are the best performance in the model family, while bold data are the best performer for the task for both embeddings. 5 random seed results averaged, with data after ± indicating standard deviations.

| Model | Epoch | Embedding | WordCt | SentCt | IsArg | ArgTyp | Coref | CoEvnt | EvntTyp2 | EvntCt | Average |
|---|---|---|---|---|---|---|---|---|---|---|---|
| TANL | 5 | FullText | $26.00_{\pm4.18}$ | $13.00_{\pm6.71}$ | $85.09_{\pm0.98}$ | $28.66_{\pm0.85}$ | $78.15_{\pm1.43}$ | $65.50_{\pm1.67}$ | $31.89_{\pm5.29}$ | $25.00_{\pm10.00}$ | 44.16 |
| | 10 | FullText | $29.00_{\pm4.18}$ | $12.00_{\pm2.74}$ | $85.47_{\pm0.52}$ | $29.47_{\pm0.63}$ | $77.41_{\pm0.88}$ | $65.50_{\pm2.30}$ | $32.70_{\pm4.72}$ | $25.00_{\pm7.91}$ | 44.57 |
| | 15 | FullText | $25.00_{\pm5.00}$ | $13.00_{\pm4.47}$ | $84.58_{\pm1.21}$ | $28.95_{\pm0.71}$ | $77.28_{\pm1.89}$ | $66.06_{\pm7.08}$ | $26.49_{\pm4.74}$ | $16.00_{\pm6.52}$ | 42.17 |
| | 20 | FullText | $25.00_{\pm7.07}$ | $15.00_{\pm11.18}$ | $83.92_{\pm1.47}$ | $28.95_{\pm0.68}$ | $77.62_{\pm0.65}$ | $65.50_{\pm5.29}$ | $31.89_{\pm2.80}$ | $22.00_{\pm4.47}$ | 43.73 |
| DyGIE++ | 5 | FullText | $18.00_{\pm7.58}$ | $14.00_{\pm8.22}$ | $81.12_{\pm1.59}$ | $30.27_{\pm0.66}$ | $73.40_{\pm0.75}$ | $60.35_{\pm5.31}$ | $32.11_{\pm3.56}$ | $18.00_{\pm9.08}$ | 40.90 |
| | 10 | FullText | $21.00_{\pm8.94}$ | $19.00_{\pm6.52}$ | $80.33_{\pm1.68}$ | $30.91_{\pm1.03}$ | $73.93_{\pm1.62}$ | $61.40_{\pm2.56}$ | $34.21_{\pm4.65}$ | $21.00_{\pm5.48}$ | 42.72 |
| | 15 | FullText | $22.00_{\pm8.37}$ | $22.00_{\pm7.58}$ | $81.07_{\pm1.27}$ | $29.97_{\pm0.36}$ | $74.57_{\pm2.26}$ | $61.40_{\pm2.56}$ | $37.63_{\pm2.88}$ | $23.00_{\pm8.37}$ | 43.96 |
| | 20 | FullText | $22.00_{\pm5.70}$ | $24.00_{\pm5.48}$ | $80.19_{\pm1.49}$ | $30.53_{\pm0.64}$ | $74.90_{\pm2.82}$ | $61.05_{\pm3.37}$ | $40.00_{\pm2.73}$ | $18.00_{\pm6.71}$ | 43.83 |
| BERT$_{base}$ | 0 | FullText | 25.00 | 20.00 | 80.84 | **31.05** | 71.07 | 63.30 | 27.40 | 20.00 | 42.33 |

**Table 8: Average Accuracy on WikiEvents Probing Task.** IE frameworks were trained over varying epochs on WikiEvents and evaluated via probing tasks on their input representations. The results, averaged over 5 trials, compare full-text embeddings against an untrained BERT baseline. Underlined figures represent the best within their model group, while bold denotes the best overall. Standard deviations are shown after ±. The performance of GTT on WikiEvents, which requires modifications for varying roles based on event type, is a topic for potential future exploration by the research community.