# OpenReview forum: "Probing Representations for Document-level Event Extraction"
_EMNLP/2023/Conference — EMNLP 2023 Findings_

### Official Review · Reviewer_FqJd · 2023-07-31

**Soundness:** 4

**Excitement:**

2: Mediocre: This paper makes marginal contributions (vs non-contemporaneous work), so I would rather not see it in the conference.

**Paper Topic And Main Contributions:**

The paper proposes using probing classifiers to understand embeddings of the encoder in the document-level information extraction tasks (IE). The authors design eight probing tasks in three categories including surface, semantic and event (task) levels. They compare four specific models in the MUC document-level IE dataset with probing classifiers, and analyze the correlations between the accuracies in probing tasks and the accuracies in the target (IE) task. The results show that the probing tasks' accuracies change during the IE task training compared to the untrained BERT-base model. They also experimented with constructing the encoder differently including different aggregation methods and different layers.

**Reasons To Accept:**

- The authors introduce probing tasks to the document-level IE tasks, and design different probing tasks for explanations.
- The probing tasks are intuitive to understand.
- The authors provide some empirical results on the correlation between probing tasks and the target task.

**Reasons To Reject:**

- The empirical results are quite mixed and hard to draw clear conclusions on the findings of explainability.
- Both the method and empirical results provide limited inspirations for future work.


**Reproducibility:**

3: Could reproduce the results with some difficulty. The settings of parameters are underspecified or subjectively determined; the training/evaluation data are not widely available.

**Reviewer Confidence:**

4: Quite sure. I tried to check the important points carefully. It's unlikely, though conceivable, that I missed something that should affect my ratings.

---

> ### Author Rebuttal · Authors · 2023-08-29
>
> We are very grateful for Reviewer FqJd’s helpful review. Firstly, we would like to add that **NLP engineering experiment** and **Reproduction study**, which our work belongs to, is also an integral part of this year’s program as indicated in the call for paper [1].
>
> We focus on understanding document-level features [2, 3] encoded in neural representations, which is an important topic but has never been studied in NLP.
>
> **Weakness 1: Mixed Results. Difficult to draw conclusions.**
>
> - We acknowledge that our empirical results seem mixed. Nevertheless, we have seen a clear differentiation between untrained encoder and IE-tuned encoder capabilities (Fig3, notice the difference between epoch 0 and 5 stays consistent as epoch number increases further). We will report averaging scores using embeddings from 5 different random states for less noisy graphs in our final version (which shows similar trends), but our conclusions mostly focus on IE-finetune v.s. Untrained. Moreover, mixed results, with claims or trends that can often be substantiated only partially or in specific parts, are often expected in explainability works [4-5].
>
> **Weakness 2: Limited Inspirations for Future Work**
>
> - We have shown a few interesting phenomena, including deteriorating coreference performance, inefficient encoding of discourse information, limited event information encoding in span encoding, etc.
>
> - Our work is the first few on probing document-level NLP tasks. Future probing work can [to-add]. Future modeling work can take insights from our work to improve on these particular metrics, and future explainability work can build on our probing framework and task design.
>
>
> **References**
>
> [1] [Call for Main Conference Papers - EMNLP 2023](https://2023.emnlp.org/calls/main_conference_papers/#contributions)
>
> [2] Document-Level Event Argument Extraction by Conditional Generation (Li et al., NAACL 2021)
>
> [3] GRIT: Generative Role-filler Transformers for Document-level Event Entity Extraction (Du et al., EACL 2021)
>
> [4] BERT Rediscovers the Classical NLP Pipeline (Tenney et al., ACL 2019)
>
> [5] Automatic Error Analysis for Document-level Information Extraction (Das et al., ACL 2022)

---

### Official Review · Reviewer_QFh8 · 2023-08-05

**Soundness:** 3

**Excitement:**

3: Ambivalent: It has merits (e.g., it reports state-of-the-art results, the idea is nice), but there are key weaknesses (e.g., it describes incremental work), and it can significantly benefit from another round of revision. However, I won't object to accepting it if my co-reviewers champion it.

**Paper Topic And Main Contributions:**

This work aims to understand event knowledge encoded in the embeddings of LLM models by employing a probing framework based on document level Information Extraction (IE) templates. The framework probes information at 3 levels, including surface-level, argument-level, and event-level. The empirical result shows that fine-tuned embeddings benefit low-abstraction-level IE tasks (e.g., surface and argument) more than high-abstraction-level tasks (e.g., event).

**Questions For The Authors:**

a) Line 202: suggest adding average sentence and document length, and their distributions to make the paper more self-explanatory. Because apparently document length can be a key factor in this study, I also suggest bucketizing  the test set by lengths to conduct the analysis. Expect to see in what length full-text contextualization cannot compete with the sentence embedding concatenation.

b)Line 218: this suggests that higher layers embed more abstract event knowledge, which is clean and nice. However, the paper does not discuss the difference between GTT and BERT and what results in the difference are not clearly discussed. Can the authors elaborate about your aspect about this and  include the discussion in the paper/Appendix?


**Reasons To Accept:**

a) Valuable evaluation result for understanding event knowledge in pre-trained LLM, which is informative to the research community.

b) Concise but clean write-up which is easy to follow.


**Reasons To Reject:**

a) The result is sort of noisy without a clean trend, though proper score aggregation could sort it out.

b) The observations can be applied to a specific dataset (MUC), but we don’t know whether it is general.


**Reproducibility:**

4: Could mostly reproduce the results, but there may be some variation because of sample variance or minor variations in their interpretation of the protocol or method.

**Reviewer Confidence:**

5: Positive that my evaluation is correct. I read the paper very carefully and I am very familiar with related work.

**Typos Grammar Style And Presentation Improvements:**

a) Line 218: Figure 4 should be mentioned in the paragraph.

b) Table 1: suggest adding average results for FullText and SentCat on the three levels of IE.

c) The observations can be applied to a specific dataset (MUC), but we don’t know whether it is general. Suggest strengthening this aspect for the paper.

---

> ### Author Rebuttal · Authors · 2023-08-29
>
> We really appreciate Reviewer QFh8’s helpful review, insightful questions, and grammar advice.
>
> **Weakness (a) Noisy results without clean trend**
>
> We acknowledge that the unaveraged result in our work seemed noisy. We have the averaged results ready, which looks only more flattened (for Fig 3) (happy to present any other part of the data during discussion). Nevertheless, the gap between the performance of BERT-base (untrained) embeddings and fine-tuned encoder’s embedding on different tasks is and stays consistent. In fact, IE performance improvements between epoch 5 and epoch20 are often only modest, which means the gap between epoch0 (BERT-base) and epoch5 in Fig 3 is the most salient and crucial to understand, and we draw most of our conclusions from this area. We will summarize these findings in the more clear textual form in the final version.
>
> **Weakness (b) Limited Dataset (MUC)**
>
> We acknowledge our observations come from limited data from a single dataset. Regrettably, document-level event datasets are scarce, and the high-quality MUC is often the one dataset people resort to or even evaluate exclusively on (See doc-level work like [1-2]). We will report limited data from WikiEvents in the appendix, but given MUC data’s higher quality (and quantity) we expect MUC results would be the most reliable.
>
> **Question (a) Text Length and Bucketizing Analysis**
>
> Thank you for your great idea. We will add a more detailed analysis to our text and appendix regarding this. As some quick facts, the average (±standard deviation) word count (entire dataset) is 362.83±220, the 25 and 75 percentile is 187 and 493, and the sentence count average is 14.15±10.
>
> **Question (b) Difference between GTT and BERT**
>
> Sorry for the possible confusion. GTT here uses BERT as its base model for encoding, and we are comparing encodings from a GTT-Finetuned BERT that was trained during the IE task training, and from the out-of-shelf BERT. We will make clear this and the interpretation of the camera-ready version.
>
> **Improvement Idea (a) Cite Fig4 on line 218 (b) Add average score per row in Table1**
>
> Great catch! We will add them.
>
> **Improvement Idea (c) Use more dataset**
>
> We will report WikiEvent results in the appendix due to the limited pages, but we prefer presenting MUC in general due to its much bigger size and richer annotation and coreference.
>
> **References**
>
> [1] Iterative Document-level Information Extraction via Imitation Learning (Chen et al., EACL 2023)
>
> [2] Template Filling with Generative Transformers (Du et al., NAACL 2021)

---

### Official Review · Reviewer_mJYE · 2023-08-12

**Soundness:** 3

**Excitement:**

3: Ambivalent: It has merits (e.g., it reports state-of-the-art results, the idea is nice), but there are key weaknesses (e.g., it describes incremental work), and it can significantly benefit from another round of revision. However, I won't object to accepting it if my co-reviewers champion it.

**Paper Topic And Main Contributions:**

The paper discusses using probing techniques to analyse the quality of the representations learnt for multiple document-level IE tasks. They claim that they are the first to explore probing in the domain of document-level IE. The main contributions of this paper include providing explainability to the performance of different document level IE models along with fine-grained analysis of the embeddings learnt for the different tasks in that domain.

**Reasons To Accept:**

The paper provides a thorough analysis about the quality of the embeddings learnt. It is one of the only works providing explainability to IE models at a document-level. The paper is well written and will also be supported with code and data, so it is a reproducible work.

**Reasons To Reject:**

While the main contribution of the paper is to explore probing in the domain of document-level IE, there is no novelty in the work as such. The results need to be presented more lucidly and interpretation is tough in some cases.

**Reproducibility:**

5: Could easily reproduce the results.

**Reviewer Confidence:**

4: Quite sure. I tried to check the important points carefully. It's unlikely, though conceivable, that I missed something that should affect my ratings.

---

> ### Author Rebuttal · Authors · 2023-08-29
>
> We are deeply thankful for Reviewer mJYE’s helpful review.
>
> **Weakness1: Novelty is not obvious as a probing work**
>
> We acknowledge that probing embedding is not new, and designing new tasks for a new problem space seems rather standard. Nevertheless, this work
>
> - focuses on understanding document-level features encoded in neural representations, which is an important topic but has never been studied in NLP.
>
> - showcases a new suite of probing tasks designed to mimic traditional pipeline approaches, with argument-level and event-level ones essential for the **document-level IE** task.
>
> - shows quantitatively and in detail how IE training improves or deteriorates the trained document encoding’s capability for the first time.
>
> - involves significant behind-the-scene engineering that could be considered novel
>
>
>   -    We annotated the MUC dataset with event triggers for compatibility, which can be utilized for document-level IE tasks later on as well [1, 2].
>
>   -   Two out of the three models are carefully enhanced for the first time to fully support generic template filling from their original (single-event-per-input, sentence-level) framework.
>
>   -   All these will be included in our codebase to be available online.
>
>
> **Weakness2: Interpretation seems tough**
>
> We acknowledge that the unaveraged result in our work seemed less straightforward. While previous probing works never explicitly mention using multiple trial results [3], we will include the averaged results of embeddings from 5 random states in our final paper (and are happy to show any part of it during our discussion). We observed very similar but smoother trends and low standard deviations across random states. We will summarize these findings in the more clear textual form in the final version and improve our graphing.
>
> **References:**
>
> [1] Document-Level Event Argument Extraction by Conditional Generation (Li et al., NAACL 2021)
>
> [2] GRIT: Generative Role-filler Transformers for Document-level Event Entity Extraction (Du et al., EACL 2021)
>
> [3] SentEval: An Evaluation Toolkit for Universal Sentence Representations (Conneau & Kiela, LREC 2018)

---

### Meta-Review · Area_Chair_5UdV · 2023-09-18

**Recommendation:** 3

**Metareview:**

The authors conduct probing on pre-trained models for document-level event extraction, covering different neural layers and different elements of structured events. Several fine-grained conclusions are drawn. This is the first attempt to conduct probing on the document level task, and novel probes that require engineering work have been designed. On the negative side, there has been some criticism on the results and the insight for guiding further research.

---

### Decision · Program_Chairs · 2023-10-07

**Decision:**

Accept-Findings

**Comment:**

The authors conduct probing on pre-trained models for document-level event extraction, covering different neural layers and different elements of structured events. Several fine-grained conclusions are drawn. This is the first attempt to conduct probing on the document level task, and novel probes that require engineering work have been designed. On the negative side, there has been some criticism on the results and the insight for guiding further research.